# Growth and Characterization of Sputtered InAlN Nanorods on Sapphire Substrates for Acetone Gas Sensing

**DOI:** 10.3390/nano14010026

**Published:** 2023-12-21

**Authors:** Ray-Hua Horng, Po-Hsiang Cho, Jui-Che Chang, Anoop Kumar Singh, Sheng-Yuan Jhang, Po-Liang Liu, Dong-Sing Wuu, Samiran Bairagi, Cheng-Hsu Chen, Kenneth Järrendahl, Ching-Lien Hsiao

**Affiliations:** 1Institute of Electronics, National Yang Ming Chiao Tung University, Hsinchu 30010, Taiwan; 2Thin Film Physics Division, Department of Physics, Chemistry, and Biology (IFM), Linköping University, SE-58183 Linköping, Sweden; 3Department of Materials Science and Engineering, National Chung Hsing University, Taichung 40227, Taiwan; 4Graduate Institute of Precision Engineering, National Chung Hsing University, Taichung 40227, Taiwan; 5Department of Applied Materials and Optoelectronic Engineering, National Chi Nan University, Nantou 54561, Taiwan; 6Department of Post-Baccalaureate Medicine, College of Medicine, National Chung Hsing University, Taichung 402010, Taiwan

**Keywords:** acetone gas sensor, InAlN, magnetron sputtering, nanorods, sapphire, DFT

## Abstract

The demand for highly sensitive and selective gas sensors has been steadily increasing, driven by applications in various fields such as environmental monitoring, healthcare, and industrial safety. In this context, ternary alloy indium aluminum nitride (InAlN) semiconductors have emerged as a promising material for gas sensing due to their unique properties and tunable material characteristics. This work focuses on the fabrication and characterization of InAlN nanorods grown on sapphire substrates using an ultra-high vacuum magnetron sputter epitaxy with precise control over indium composition and explores their potential for acetone-gas-sensing applications. Various characterization techniques, including XRD, SEM, and TEM, demonstrate the structural and morphological insights of InAlN nanorods, making them suitable for gas-sensing applications. To evaluate the gas-sensing performance of the InAlN nanorods, acetone was chosen as a target analyte due to its relevance in medical diagnostics and industrial processes. The results reveal that the InAlN nanorods exhibit a remarkable sensor response of 2.33% at 600 ppm acetone gas concentration at an operating temperature of 350 °C, with a rapid response time of 18 s. Their high sensor response and rapid response make InAlN a viable candidate for use in medical diagnostics, industrial safety, and environmental monitoring.

## 1. Introduction

Ternary alloys are a class of materials that consist of three distinct elements, carefully combined to achieve specific properties and characteristics tailored for various industrial applications [1,2,3,4]. These alloys represent a fascinating realm of materials science, offering a unique balance of properties that can be finely tuned through the precise control of composition. In this context, indium aluminum nitride (InAlN) alloys have garnered significant attention in recent years due to their exceptional electronic, optical, and thermal properties, making them a promising candidate for a wide range of advanced technologies and devices [5,6,7,8]. It is well known that organic solvents are often used in industries. These common organic solvents include isopropyl alcohol and acetone, which are used for processes such as wafer cleaning [9] and photoresist removal [10]. Acetone is considered a volatile organic compound with similar characteristics to other organic volatile gases such as benzene, ethanol, formaldehyde, etc., and excessive inhalation can lead to neurological toxicity [11,12]. Additionally, acetone is a common volatile organic compound in human exhaled breath [13,14]. The concentration of acetone in exhaled breath correlates strongly with glucose and acetone concentrations in blood [15,16]. The concentration of acetone in exhaled breath is often considered as a biomarker for diabetes and kidney disease [17]. According to the regulations of the Occupational Safety and Health Research Institute, the permissible concentration standard for acetone, known as the threshold limit value, is 937.5 ppm for short-term exposure and there is a recommended 8 h time-weighted average concentration of 500 ppm [18,19]. Semiconductor materials find extensive applications in gas sensors. Among various semiconductor materials, InAlN with properties such as high temperature resistance [20], high pressure resistance [21], low power consumption [22], and low resistance [23] are highly favorable. Semiconductor gas sensors offer irreplaceable advantages in terms of low cost, compact size, rapid response, low power consumption, and improved sensitivity, making them suitable for home testing and personal use [24,25,26]. The basic sensing principle of semiconductor gas sensors is straightforward. The sensing epilayer reacts with the target gas on its surface, causing a change in the fundamental electrical properties of the sensor element. By measuring the change in these electrical properties and performing simple calculations, the sensor provides a numerical value as an indicator of the target gas concentration. Various semiconductor oxide materials have been explored for the development of acetone gas sensors. Noteworthy examples include the work by Jang et al., who reported on a sensor based on SnO_2_ nanotubes. Their sensor exhibited a response of 18.4% at an operating temperature of 350 °C, albeit with a relatively extended response time exceeding 120 s [27]. Similarly, Choi et al. presented WO_3_ nanofiber-based acetone gas sensors, achieving a remarkable sensor response of 121.5% at an operating temperature of 350 °C, though with a response time of around 32 s [28]. Addressing the need for improved response times, Koo et al. reported on an Al-doped ZnO-based acetone gas sensor with a quick response time of 2.9 s. However, it is worth noting that the operating temperature for this sensor was relatively high, reaching 450 °C [29]. In light of these findings, our study focuses on utilizing InAlN nanostructured films (nanorods) as a promising alternative, demonstrating a quick response time to acetone gas. The epitaxial growth of InAlN films using the magnetron sputtering method is cost-effective and shows significant potential for application in industrial-scale markets due to its affordability [30,31]. This work addresses a significant research gap, as while there have been considerable efforts devoted to depositing and characterizing InAlN films and nanostructures on various substrates [31,32,33,34,35], relatively limited attention has been directed toward harnessing their potential as acetone gas sensors.

The aim of this work is to demonstrate the growth and characterization of InAlN nanorods on sapphire substrates using magnetron sputter epitaxy (MSE) with different indium compositions. According to our knowledge, these films are not employed in acetone gas sensors. Therefore, this work demonstrates the gas-sensing properties, including sensor response, response time, and recovery time, under controlled conditions. In addition to this, a theoretical model employing the density functional theory (DFT) generalized gradient approximation (GGA) method has been introduced to elucidate the gas-sensing mechanism of acetone on an InAlN sensor. The abundant dense nanorods contribute to facilitating efficient gas adsorption and desorption processes, making them a promising candidate for developing sensitive and responsive gas sensors.

## 2. Materials and Methods

The growth of InAlN nanostructured films (235 nm thick) was performed on single-crystal *c*-plane (0001) sapphire substrates using ultra-high vacuum MSE [34,35,36]. Substrates underwent a rigorous cleaning process involving immersion in acetone (ACE) and isopropanol (IPA) for 5 min each, with ultrasonic agitation prior to the deposition of films. Nitrogen (N_2_) gas was used to blow away water droplets from the cleaned substrates. The cleaned substrate was placed inside the sputtering chamber, which was evacuated to a background pressure of 5 × 10^−9^ Torr. Two sputtering targets were employed: a 2-inch In target (purity: 99.999%) and a 3-inch Al target (purity: 99.999%). A mixture of argon and N_2_ gases was introduced into the chamber, and the substrate temperature was set to 300 °C. Magnetron sputtering of the In target was carried out at a fixed power of 10 W. Magnetron sputtering of the Al target was conducted with variable powers of 30 W, 150 W, and 300 W to grow InAlN films with different compositions. Figure 1 depicts several key steps performed in the growth and fabrication of the gas sensor device. First, the sapphire substrate was cleaned by ACE and IPA, shown in Figure 1a. Then, the InAlN layer was deposited by MSE. After this, the sensing area was defined by a photoresistor (PR) and etching by an induced coupled plasma reactive ion etching (ICPRIE) system, shown in Figure 1c,d. In the following step, the electrode area was defined by PR, the Ti/Al/Ni (20/300/25 nm) multilayers were deposited on the sensing epilayer as an ohmic contact, and the PR was lifted off, shown in Figure 1e–g. The final gas sensor was examined using an optical microscope and is shown in Figure 1h. The output signals and readings were recorded using a B1505A Keysight power device analyzer. The power device analyzer was set to give output at a stable voltage of 1 V for the gas sensor to monitor the currents.

## 3. Results

Figure 2 presents X-ray diffraction (XRD) patterns obtained from InAlN films grown on sapphire substrates, revealing valuable insights into the structural characteristics of these materials. The observed diffraction peaks have been successfully indexed using the JCPDS card No. 65-3412 for InN and JCPDS card No. 653409 for AlN. Notably, the diffraction patterns from all three samples exhibit a positioning of peaks between the characteristic diffraction peaks of InN (0002) and AlN (0002) at 2θ angles of 31.34° and 36.04°, respectively. This crucial observation strongly suggests the growth of InAlN films along the *c* axis. With increasing Al magnetron power, the diffraction peaks shift towards higher 2θ angles, implying a decrease in In composition (x) in the In_x_Al_1−x_N films, where the composition is estimated using Vegard’s law determined by the *c* lattice constant of In_x_Al_1−x_N calculated by Bragg’s law [36]. These findings align with prior literature, where In_x_Al_1−x_N ternary compounds have been grown on sapphire substrates with In composition falling within the range of 0.35–0.68, particularly for applications in photodetectors [36].

Figure 3 depicts the surface roughness of InAlN films and present the morphology of surface nanostructures in 3D images using atomic force micrographs. The films on the three specimens have different In compositions of 29%, 54%, and 93%. The surface roughness values for these films are 2.91 nm, 1.57 nm, and 2.74 nm, respectively. Moreover, the surface morphologies also indicated that the InAlN material structures do not seem to be a compact film but nanorods.

Figure 4a depicts a top-view SEM micrograph of InAlN film with 93% In composition (In_0.93_Al_0.07_N), which presents a nanostructure, and the diameter is in the range of 30–60 nm. The obtained surface morphology is consistent with that obtained by AFM shown in Figure 3. The cross-sectional transmission electron microscopy (TEM) micrograph is shown in Figure 4b, which provided crucial insights into the structural characteristics of the InAlN film. Specifically, it has revealed the presence of nanorods, which were aligned vertically on the sapphire substrates with a film thickness of 235 nm. The characteristics observed in the SEM and TEM micrographs suggest that this material is well suited for gas-sensing applications due to its nanostructural morphology. The small size and one-dimensional nature of nanorods can facilitate the diffusion, absorption, and desorption of gas molecules through the material. This could allow gas molecules to penetrate deeper to cover the entire nanorod surface, increasing the chances of interactions with active sites and resulting in faster response times. Therefore, such a nanostructure consisting of abundant dense nanorods is expected to facilitate efficient gas adsorption and desorption processes, making it a promising candidate for developing sensitive and responsive gas sensors.

Figure 5 depicts the gas-sensing characteristics of the InAlN-nanorod-based acetone gas sensor, operating at 350 °C with a DC bias of 1 V. After a 30 min stabilization period, the responses of specimens with varying In compositions to different acetone gas concentrations were measured. The real-time variations in current values were meticulously recorded, and the results are presented in Figure 5. The schematic diagram of the device measurement setup is shown in Figure 5a. In context of the nanostructured film with a low In composition of 29% (In_0.29_Al_0.71_N), shown in Figure 5b, a remarkable observation was made: there was an absence of any discernible change in current values, irrespective of the introduction of varying concentrations of acetone gas or the return to ambient air. This intriguing finding suggests an extraordinary insensitivity of the nanorods to acetone gas. It could be possible that the In_0.29_Al_0.71_N film falls outside this optimal range for acetone sensing. Figure 5c,d represent the acetone-gas-sensing characteristics of In_0.54_Al_0.46_N and In_0.93_Al_0.07_N films, revealed an interesting responsive profile upon introduction and removal of acetone gas.

Figure 6a depicts the sensor response of In_0.54_Al_0.46_N and In_0.93_Al_0.07_N nanostructured films. The gas sensor response, *S* (%), is defined as follows:(1)$$S\;(\%)=(I_{g}-I_{a})/I_{g}\times 100$$
where *S* (%) is the gas sensor response in percentage, *I_g_* is the current across the sensor upon exposure to the analyte, and *I_a_* is the current across the sensor when exposed to dry air. Since low-indium-composition In_0.29_Al_0.71_N film is not found to be responsive to acetone gas, it is excluded from any further analysis. Upon exposure to different concentrations of 600, 150, and 30 ppm of acetone gas, the In_0.54_Al_0.46_N sensor exhibited response percentages of 5.79%, 0.71%, and 0.62%, respectively. In the context of In_0.93_Al_0.07_N nanorods, the sensor exhibited response percentages of 2.33%, 1.35%, and 0.70% when exposed to varying concentrations of 600, 150, and 30 ppm of acetone gas. Figure 6 reveals intriguing nuances in the gas-sensing capabilities of these InAlN nanorods. Notably, the In_0.54_Al_0.46_N nanorods exhibited superior gas-sensing response when exposed to 600 ppm acetone gas, outperforming their counterparts. However, the scenario changes when the concentration is lowered to 150 ppm, where the In_0.93_Al_0.07_N nanorods emerge as the more effective sensor. This observed phenomenon can be attributed to the response percentages of both films, which remain below the 10% threshold. Figure 6b,c present the transient response of the annealed InAlN films to varying acetone gas concentrations. The transient response of a gas sensor refers to how quickly and effectively the sensor reacts to changes in gas concentration or environmental conditions. The ability of sensors to detect and adapt to variations in gas levels is typically measured from an initial baseline state to a new equilibrium state. The response time and recovery time are the key characteristics of a gas sensor. The response time is the time it takes for the sensor’s output to change from its initial value to 90% of its final equilibrium value after exposure to a gas or change in conditions. The recovery time is the time it takes for the sensor’s output to return to its baseline or pre-exposure level. Upon close examination of Figure 6b,c, a notable trend becomes evident. As the concentration of introduced acetone gas increases, both the response time and recovery time exhibit an upward trajectory. Furthermore, the In_0.93_Al_0.07_N gas sensor exhibited a fast response time of 18 s and quick recovery time of 120 s, achieving equilibrium significantly faster than the In_0.54_Al_0.46_N sensor (recovery time 399 s). This observation highlights the pivotal role of epilayer composition in determining the sensor’s behavior.

Since InAlN is an *n*-type semiconductor and acetone gas is a reducing gas, a reduction in depletion width layers is favorable upon exposure to reducing gas. In recent studies, we have successfully illustrated the electron flow from the reducing gas to the *n*-type semiconductor until the Fermi level equilibrium is achieved [37,38,39]. This phenomenon leads to the formation of ohmic contact. An ohmic contact denotes a junction between the reducing gas and the semiconductor where current can flow bidirectionally. This ohmic contact facilitates the efficient migration of electrons from within the semiconductor to its surface, subsequently diminishing the surface resistance of the semiconductor. Figure 7 depicts the proposed acetone-gas-sensing mechanism for the InAlN gas sensor. When InAlN encounters the atmosphere, oxygen molecules are adsorbed onto its surface, leading to the extraction of electrons from the conduction band as shown in Figure 7a. This electron transfer results in the generation of ions through the trapping of electrons. Due to the bandgap of In_0.93_Al_0.07_N being smaller than that of In_0.54_Al_0.46_N, In_0.93_Al_0.07_N easily traps electrons before sensing. This can be demonstrated by the current of In_0.93_Al_0.07_N being higher than that of In_0.54_Al_0.46_N, shown in Figure 5c,d. Consequently, a reduction in electron density near the surface gives rise to the formation of an electron depletion layer (EDL) and the establishment of a barrier potential. The specific chemisorbed oxygen ions (O^2−^ or O^−^) that form is contingent upon the operating temperature of the gas sensors.
O_2(gas)_ + e^−^ → O^2−^_(ads)_ (T ≤ 150 °C)(2)
O^2−^_(ads)_ + e^−^ → 2O^−^_(ads)_ (150 °C ≤ T ≤ 300 °C)(3)
O^−^_(ads)_ + e^−^ → O^2−^_(ads)_ (T ≥ 300 °C)(4)

However, at operating temperatures surpassing 150 °C, the predominant interaction involves the adsorption of O^−^ ions on the InAlN sensor surface. Upon exposure to the acetone gas, the sensor’s surface absorbs gas molecules, which subsequently react with the adsorbed oxygen ions. The resultant change in resistance depends on the majority carriers in the InAlN nanorods. Specifically, for the reaction of acetone gas with adsorbed oxygen ions, the following equation describes the process: CH_3_COCH_3 (ads)_ + 8O^δ−^_(ads)_ → 3CO_2 (gas)_ + 3H_2_O _(vap)_ + 8e^δ−^(5)
where δ represents the empirical constant, commonly denoted as integers 1 and 2 for O^−^ and O^2−^ adsorbed oxygen species, representing the typical values used to describe these adsorbed oxygen species [40,41]. This gas–solid interaction and its influence on sensor response underscore the crucial role of surface chemistry in gas-sensing applications. The transfer of electrons between the acetone gas and the surface of InAlN nanorods plays a pivotal role in altering the width of the EDL. This electron transfer mechanism results in a noticeable alteration in the overall resistance of InAlN-based sensors. For instance, when acetone vapor is introduced into the sensor chamber and interacts as depicted in Figure 7b, the oxygen species react with acetone gas molecules and releases the trapped electrons back to the InAlN, leading to a reduction in the EDL width and, consequently, a decrease in sensor resistance. It is known that the electron affinity of In_0.93_Al_0.07_N is higher than that of In_0.54_Al_0.46_N, which could dominate the electron transfer from acetone to the semiconductor. It is noteworthy that the InAlN sensor exhibits a sensing range spanning from 1 to 600 ppm, operates at 350 °C, and demonstrates a responsivity of 2.33% at 600 ppm, with a response time of 18 s and a recovery time of 120 s, which are significant upon comparison with the literature. These performance characteristics evaluated above show that the InAlN nanorods are suitable for specific gas-sensing applications.

According to above experiments, the gas sensitivity of In_0.93_Al_0.07_N sensors is more effective when the concentration of acetone vapor is reduced to 150 ppm. Therefore, we believe that small differences in Al content have a greater impact on gas sensitivity. To gain more insight into the acetone gas sensitivity of InAlN films from an atomistic viewpoint, we constructed geometric models of InN, In_0.97_Al_0.03_N-1, In_0.97_Al_0.03_N-2, and In_0.5_Al_0.5_N, which are terminated by In or Al atoms. The labels “1” and “2” in the models represent reactants that introduce substituted Al atoms directly into the In sites of InN lattices in the first and second layers close to the surface. In this study, In_0.97_Al_0.03_N represents an alloy where one In atom is replaced by an Al atom in an InN alloy initially comprising 36 In atoms and 36 N atoms. To quantify the alteration in atomic composition, the proportions of In and Al atoms were calculated. This was achieved by dividing the number of In (or Al) atoms by the total number of metallic atoms in the alloy. Consequently, this calculation reveals that the approximate atomic ratio of In to Al in the modified alloy is 0.97:0.03. The models are optimized in detail using standard density functional theory (DFT) methods, employing the generalized gradient approximation (GGA) with the Perdew–Wang (PW91) correction within the Vienna Ab Initio Simulation Package (VASP) in all cases [42,43,44,45]. We have employed repeated slab geometries with 72-atom InAlN(111) films with in-plane lattice constants of 10.75 Å × 10.75 Å and 10.02 Å × 10.02 Å, which were constructed using bulk crystalline configurations of bulks InN and In_0.5_Al_0.5_N (space group: 186 *P*63*mc*), respectively. These configurations were chosen to sufficiently decouple the interactions between acetone molecules, with a vacuum separation equivalent to 40 Å to prevent interactions between the top and bottom of the slab. A plane-wave cutoff energy and the self-consistent total energy criterion were set to 450 eV and 10^−4^ eV/atom, respectively, along with a 3 × 3 × 1 gamma-centered *k*-point grid. It is important to note that the work function (Φ_S_) is defined by the following equation:(6)$$\Phi_{S}=E_{\mathrm{VAC}}-E_{F}$$
*E_VAC_* and *E_F_* represent the vacuum energy level and the Fermi level, respectively. The vacuum energy level is determined by the vacuum position energy, calculated using the planar and macroscopic average method for the electrostatic potential, along the direction perpendicular to the heterojunction interface. The optimized atomic structures with adsorbed acetone molecules denoted by InN-CH_3_COCH_3_, In_0.97_Al_0.03_N-1-CH_3_COCH_3_, In_0.97_Al_0.03_N-2-CH_3_COCH_3_, and In_0.5_Al_0.5_N-CH_3_COCH_3_ are as shown in Figure 8. Furthermore, the energies of the vacuum level and Fermi level, work function, and work function change in our geometric models are shown in Table 1. In our previous studies [37,38,39], the alterations in the work function of CO, NO_2_, and H_2_S molecules upon interaction with the ZnGa_2_O_4_(111) surface align with experimental findings. These changes in the work function signify the overcoming of electron barriers induced by gas adsorption, contributing to the quantitative assessment of gas-sensing characteristics in zinc gallium oxide. The work function changes serve as indicators of the quantified sensitivity achievable in ZnGa_2_O_4_-based gas sensors. When an acetone molecule is adsorbed onto the surface atom of Al on the In_0.97_Al_0.03_N(0001) surface, the most substantial alteration in work function change is observed, with a significant decrease of −1.19 eV. This observation shows the remarkable capability of the surface atom Al on the InN(0001) surface to notably enhance gas sensitivity or induce substantial work function variations. In contrast, when Al substitution occurs in the second layer, further away from the surface, the corresponding work function changes are noticeably diminished, with a decrease of −0.83 eV. A small amount of Al in InN may be randomly distributed within the bulk or on the surface. According to DFT calculations, the distribution of Al on the surface yields the most significant alteration in the work function. Experimental data also indicate an enhanced gas-sensing sensitivity in the presence of trace amounts of Al. This suggests that the distribution of trace Al predominantly occurs on the surface rather than within the bulk material. This phenomenon can be rationalized from a thermodynamic perspective, indicating a propensity for point defects to migrate to the surface, reducing the system’s energy. This is because that the surface is known as the locus of the material’s most substantial defects. However, in the case of In_0.5_Al_0.5_N(0001) surfaces, composed of surface atoms containing an equal combination of 50% Al and 50% In, the work function change resulting from acetone molecule adsorption remains slightly lower than that observed when only a single surface atom Al is present on the In_0.97_Al_0.03_N(0001) surface. This result illustrates our experimental findings that when the concentration of acetone gas is reduced to 150 ppm, nanorods with an In composition of 93% prove to be the more efficient sensor. When the acetone gas concentration exceeds 200 ppm, nanorods comprising 50% In or Al exhibit a greater number of surface Al atoms, serving as enhanced ohmic contact points. Consequently, it is not surprising that enhancing the number of ohmic contact points amplifies the gas-sensing response.

## 4. Conclusions

This work has presented the successful growth and characterization of ternary alloy InAlN with varying indium compositions using magnetron sputter epitaxy for potential applications in acetone gas sensors. The systematic investigation of the structural and morphological properties of the InAlN nanorods has provided valuable insights into their suitability for gas-sensing applications. These findings indicate that the InAlN nanorods exhibit promising attributes, including a well-defined growth of nanorods along the *c* axis and excellent structural integrity. Furthermore, the growth process allows for precise control over In composition, which is a crucial factor for enhancing gas-sensing performance. The sensing mechanism, which reveals the sensitivity to InAlN at varying Al concentrations, has been elucidated through DFT-GGA modeling and calculations. The results also underscore the significance of surface Al atoms serving as enhanced ohmic contact points. The gas-sensing experiments conducted in this work demonstrated the InAlN nanorods’ sensor response of 2.33% to acetone gas, highlighting their potential as a reliable sensor platform for detecting acetone, a significant biomarker for various medical and industrial applications. The response time of 18 s and recovery time of 120 s towards 600 ppm acetone gas concentration observed in this work indicate that InAlN nanorods can provide rapid and accurate detection of acetone gas, which is critical for real-time monitoring and control.

## Figures and Tables

**Figure 1 nanomaterials-14-00026-f001:**
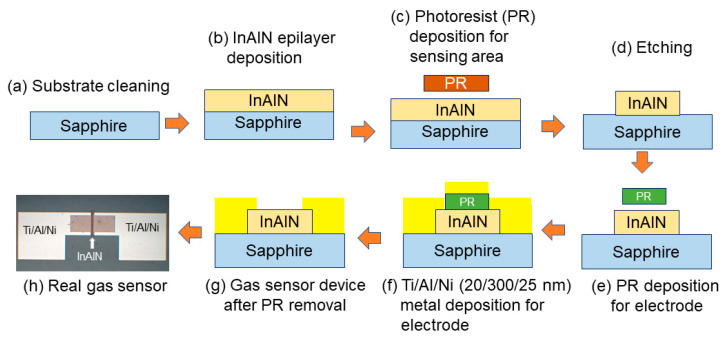
Fabrication process of InAlN gas sensor.

**Figure 2 nanomaterials-14-00026-f002:**
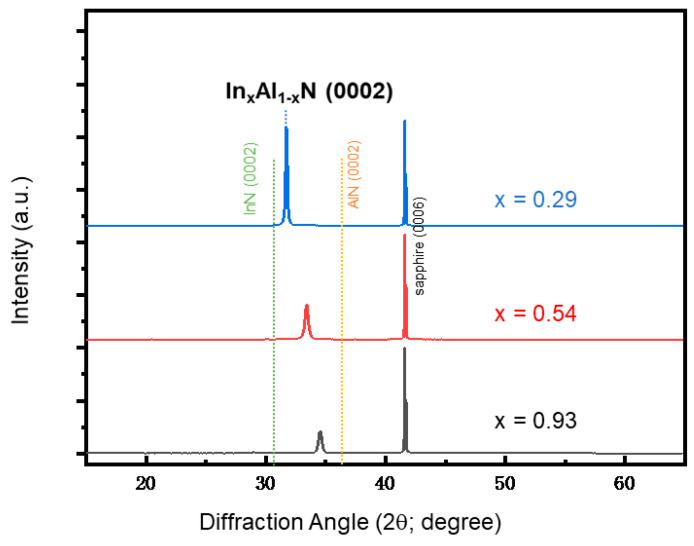
XRD patterns of InAlN films with different In compositions.

**Figure 3 nanomaterials-14-00026-f003:**
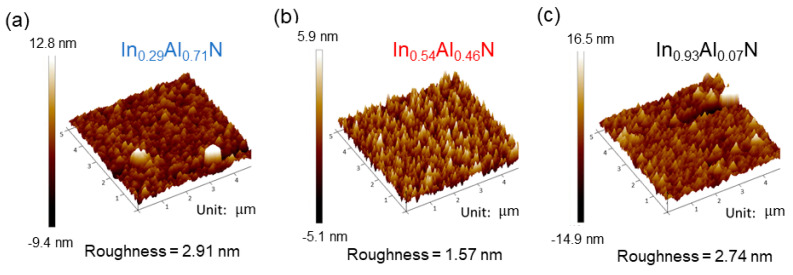
Atomic force micrographs of InAlN films with (**a**) 29%, (**b**) 54%, and (**c**) 93% In compositions.

**Figure 4 nanomaterials-14-00026-f004:**
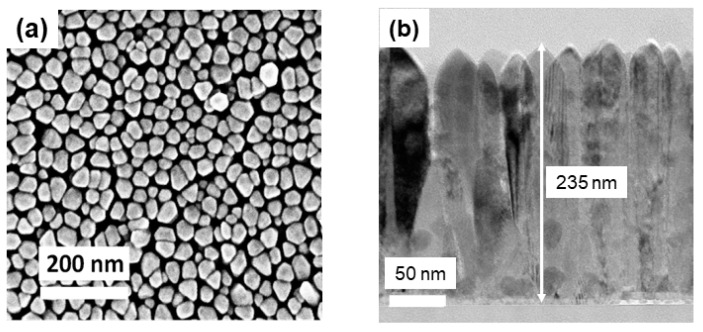
(**a**) Top-view SEM micrograph and (**b**) cross-sectional TEM micrograph of In_0.93_Al_0.07_N nanorods.

**Figure 5 nanomaterials-14-00026-f005:**
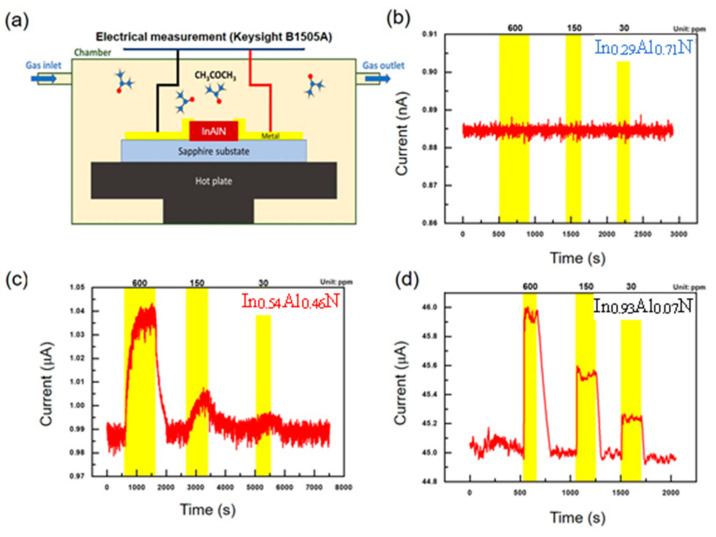
Gas-sensing characteristics of InAlN nanostructured films with different acetone gas concentrations: (**a**) schematic diagram of the measurement system, (**b**) In_0.29_Al_0.71_N, (**c**) In_0.54_Al_0.46_N, and (**d**) In_0.93_Al_0.07_N. Operation temperature of these sensors was set at 350 °C. Noted that the yellow color ranges related to inject gas concentration shown in the above x-axials.

**Figure 6 nanomaterials-14-00026-f006:**
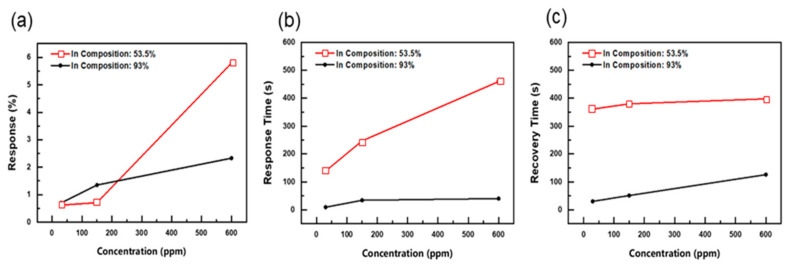
(**a**) Sensor response, (**b**) response time, and (**c**) recovery time for In_0.54_Al_0.46_N and In_0.93_Al_0.07_N gas sensors.

**Figure 7 nanomaterials-14-00026-f007:**
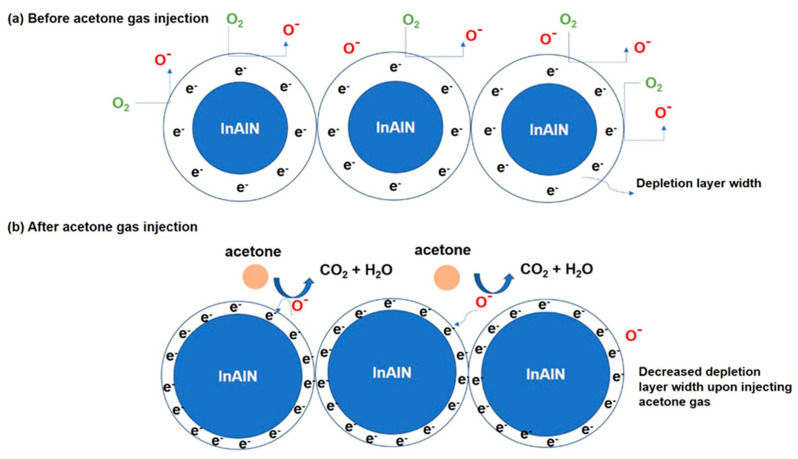
Proposed gas-sensing mechanism of InAlN-based acetone gas sensor.

**Figure 8 nanomaterials-14-00026-f008:**
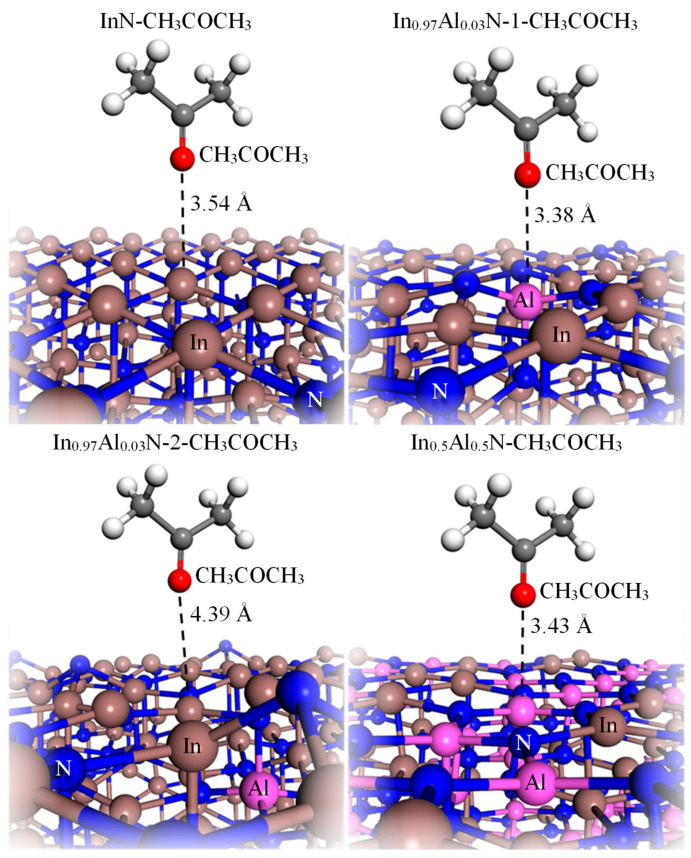
Ball and stick structural representations of the optimized InN-CH_3_COCH_3_, In_0.97_Al_0.03_N-1-CH_3_COCH_3_, In_0.97_Al_0.03_N-2-CH_3_COCH_3_, and In_0.5_Al_0.5_N-CH_3_COCH_3_ considered in this study. Atoms are represented by spheres: In (brown), Al (pink), N (blue), C (gray), O (red), and H (white). Bond lengths are given in Å.

**Table 1 nanomaterials-14-00026-t001:** Calculated vacuum energy *E_VAC_*, Fermi energy *E_F_*, work function without acetone molecules Φ_S_ and with acetone molecules Φ_S_, gas, and work function changes ΔΦ of InN-CH_3_COCH_3_, In_0.97_Al_0.03_N-1-CH_3_COCH_3_, In_0.97_Al_0.03_N-2-CH_3_COCH_3_, and In_0.5_Al_0.5_N-CH_3_COCH_3_. All energies are presented in eV.

Model	*E_VAC_* (eV)	*E_F_* (eV)	Φ_S, gas_ (eV)	Φ_S_ (eV)	ΔΦ (eV)
InN	0.15	−2.47	-	2.62	-
In_0.97_Al_0.03_N-1	1.67	−2.62	-	4.29	-
In_0.97_Al_0.03_N-2	0.29	−2.34	-	2.63	-
In_0.5_Al_0.5_N	1.64	−2.85	-	4.49	-
InN-CH_3_COCH_3_	0.53	−1.78	2.31	-	−0.31
In_0.97_Al_0.03_N-1-CH_3_COCH_3_	1.03	−2.07	3.10	-	−1.19
In_0.97_Al_0.03_N-2-CH_3_COCH_3_	0.09	−1.71	1.80	-	−0.83
In_0.5_Al_0.5_N-CH_3_COCH_3_	0.96	−2.37	3.33	-	−1.16

## Data Availability

The data presented in this study are available on request from the corresponding author.

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
