# Peer review of "Growth and Characterization of Sputtered InAlN Nanorods on Sapphire Substrates for Acetone Gas Sensing"

_nanomaterials, 2023, doi:10.3390/nano14010026_

Round 1

Reviewer 1 Report

Comments and Suggestions for Authors

The manuscript reports results of an experimental study on the fabrication and characterization of nanorods of a ternary alloy indium aluminum nitride (In_x Al_(1-x) N, x=0.93, 0.54, and 0.29) n-type semiconductors, grown on sapphire substrates using an ultra-high vacuum magnetron sputter epitaxy.  The paper also reports results of solid state DFT calculations simulating the adsorption of acetone on the InAlN film.  The manuscript explores the potential of such materials for acetone gas sensing applications, given its use in medical diagnostics and various industrial processes.

The main claim of the manuscript is that the rapid response time (18 s) of the sensor under normal operation conditions make InAlN a viable candidate for use in medical diagnostics industrial safety, and environmental monitoring.

The key hypothesis of the study is that abundant dense nanorods facilitate the adsorption of gas molecules (higher surface to volume ratio) and diffusion of charge through the material, contributing to increased sensing sensitivity. 

Overall, the manuscript is very concise, to the extent that it lacks some useful information:

1. The introduction does not provide sufficient background, particularly regarding the state of the art in acetone sensing.  To put the present results in context the manuscript needs to report the sensitivities and response times of present sensing materials.

2. The manuscript claims that the gas sensitivity could be related with the work function change and provides several references [26–28] on line 267.  However, a few details and a short discussion would benefit the reader.

3. The correlation between the experimental results and the DFT simulation is of high interest but the discussion in the text is very brief.  It is interesting that the Al sites cause larger change in the work function than the In sites, particularly at low Al composition (high x).  However, at high x, the few existing Al sites may also be located in the bulk, not at the surface (case 2), in which case the change in work function is small, close to the one observed for InAl.  On the other hand, the manuscript claims that the increase in composition towards parity (x=50%) improves performance by decreasing the series resistance of the sensor.  Under these circumstances, what is the optimum composition between the two opposing tendencies?  The issue needs to be more thoroughly discussed and measurements at an intermediate composition, with x lower than 0.97 but higher than 0.55 may be of high interest.

Overall, the manuscript is interesting but needs to address the concerns raised to reach the high standards of Nanomaterials.

Comments on the Quality of English Language

Only minor editing of English language is needed. 

Author Response

Please refer the attached file.

Reviewer 2 Report

Comments and Suggestions for Authors

The manuscript entitled: Growth and Characterization of Sputtered InAlN Nanorods on Sapphire Substrates for Acetone Gas Sensing is of great interest for developments in gas sensing, but before being published, some shortcomings should be addressed:

Introduction:

Please provide specific references to:

1. In a wet bench environment, common organic solvents include isopropyl alcohol and acetone, which are used for processes such as wafer cleaning [xxx]  and photoresist removal [xxx].

AND

2. Among various semiconductor materials, InAlN with properties such as high temperature resistance [xxx], high pressure resistance [xxx], low power consumption [xxx], low resistance [xxx] are highly favorable.

3. The main targets are not appropriately clearly presented. Please re-write the aims of this work. 

4. Did you thy to decrease the operating temperature? Is such a high temperature a demand ?

5 Can you presume why the differences in sensitivity regarding  In0.54Al0.46N and  In0.93Al0.07N?

6. Introduction of few Sub-Chapters would make the understanding more easier. (sensor characterization, presumed mechanism for sensing function of temperature...influence of Al contain)

Author Response

Please refer the attached file.

Round 2

Reviewer 1 Report

Comments and Suggestions for Authors

The revised manuscript has addressed two of my concerns and has improved accordingly.  However, my third comment indicated that the manuscript has a gap between the compositions with x between 0.55 and 0.97, such that the optimum alloy is not reported.  That was the reason I suggested that measurements at an intermediate composition are performed.

In their response, the authors stated that ‘investigations into compositions ranging between In contents of 0.97 and 0.50 extend beyond the scope of this study due to their high complexity and structural considerations’, in the context of difficulties with calculations of random samples.  Obviously, the response ignores the suggestion to perform more experiments (not calculations!), which was the main reason to recommend a major revision. 

More importantly, the response suggests that finding the optimum composition of the ternary alloy is an investigation beyond the scope of the study.  Such suggestion is in contradiction with the main claim of the manuscript that InxAl1-xN is a viable candidate for use for acetone sensing in medical diagnostics industrial safety, and environmental monitoring.  

I continue to believe that the manuscript is interesting and valuable but, in the present form, it seems incomplete, as it does not indicate the range of compositions that optimize acetone sensing.  Under these circumstances, it does not fully serve its claimed purpose.

Author Response

In their response, the authors stated that ‘investigations into compositions ranging between In contents of 0.97 and 0.50 extend beyond the scope of this study due to their high complexity and structural considerations’, in the context of difficulties with calculations of random samples. Obviously, the response ignores the suggestion to perform more experiments (not calculations!), which was the main reason to recommend a major revision.

More importantly, the response suggests that finding the optimum composition of the ternary alloy is an investigation beyond the scope of the study.  Such suggestion is in contradiction with the main claim of the manuscript that InxAl1-xN is a viable candidate for use for acetone sensing in medical diagnostics industrial safety, and environmental monitoring.  

Resp.: Thank reviewer’s comment. In this manuscript, the quick response was found in the high In composition (In 0.97Al0.03N). As In decreasing, the response time became obviously long. From the tendency, the short response time between the compositions with x between 0.55 and 0.97 could not be obtained. Nevertheless, the performance of InxAl1-xN gas sensor with x between 0.55 and 0.97 is important and is under study. More details calculation and experiments about the gas sensors made of InxAl1-xN with x between 0.55 and 0.97 will be discussed and submitted in our next papers.
